# Effectiveness of Prolonged Application of Super High-Intensity Continuous Training—Team Case Study

**DOI:** 10.3390/jfmk10030241

**Published:** 2025-06-25

**Authors:** Miloš M. Milošević, Jovana Popović, Milivoj Dopsaj, Milenko B. Milosević

**Affiliations:** 1Faculty of Physical Education and Sports Management, Singidunum University, Danijelova 32, 11000 Belgrade, Serbia; 2Community Health Centre Rakovica, 11000 Belgrade, Serbia; dragajovana@gmail.com; 3Faculty of Sport and Physical Education, University in Belgrade, 11000 Belgrade, Serbia; milivoj.dopsaj@gmail.com; 4Serbia Center for Diagnostics and Training Design CEDIP, 11000 Belgrade, Serbia; mlsvc2010@gmail.com

**Keywords:** aerobic profile, VO_2_max, team handball, women’s sports

## Abstract

**Background:** Super High-Intensity Continuous Training (SHCT) is a type of aerobic training program that combines high intensity with continuous loads, such as running for 20 min at 75%, 80%, or even 95% of the velocity at maximal oxygen uptake. Recent studies show significant positive effects, but the consequences of prolonged use remain unknown. **Purpose:** This study aims to investigate and evaluate the effects of prolonged application of the SHCT model in elite team handball players. **Method:** For this purpose, a field-based quasi-experiment was organized using the SHCT training model on 14 professional female team handball players competing in the first national league who participated in 16 weeks of SHCT training during the competition season. **Results:** After the application of SHCT training, the increases in the parameters of the aerobic profile (distance run in Cooper’s 12 min run test, maximum rate of oxygen consumption, value of the maximum relative oxygen consumption, running speed for which maximum rate of oxygen consumption occurs) reached from 25.4% to 35.2%. The effect size of these changes was η^2^p > 0.90 and was significant at the *p* < 0.001 level. **Conclusions:** The investigated aerobic model is effective. Therefore, its use is recommended for designing aerobic training for elite teams and the general sports population.

## 1. Introduction

The modern female team handball game demands that players endure alternating aerobic and anaerobic loads. This is often accompanied by shorter and longer breaks, with frequent, sudden, and rapid changes in both intensity and direction of movement during gameplay [1,2,3]. In such a structure, aerobic effort prevails during matches [4,5]. Namely, at the top competitive level, the average load is 79.4 ± 6.4% of the maximum rate of oxygen consumption (VO_2_max) for a full game [4]. At the same time, the results of studies with laboratory and field tests show that the maximum relative oxygen consumption (VO_2_rel) of the players from various national teams ranges from 45 to 58 mL·kg^−1^·min^−1^ [5]. A recent study that analyzes team handball on-court physical performance in elite, top-elite, and world-class female players finds significant differences among groups in aerobic parameters (elite VO_2_rel = 54.2 ± 2.9 mL·kg^−1^·min^−1^; top-elite VO_2_rel = 60.6 ± 4.8 mL·kg^−1^·min^−1^; and world-class VO_2_rel = 64.3 ± 6.4 mL·kg^−1^·min^−1^) [6]. The ability of players to fulfill the demands of the game at the top international level, which constantly requires new and more efficient training models, has increased in recent years [7,8]. The aim is to improve aerobic characteristics as well.

Several models exist for designing aerobic training, which can be classified into two categories: continuous and interval training. Continuous training enhances capillarization and the oxidative capacity of muscles [9]. In contrast, interval training primarily improves the strength of the heart muscles by incorporating varying intensities, which boosts circulation and enhances the delivery of oxygen to the muscles [10]. The objective of both models is to increase aerobic capacity by positively influencing both the organic and metabolic systems. With the aim to meet growing demands, Super High-Intensity Continuous Training (SHCT), which combines high intensity with continuous load, is proposed [11]. This model significantly differs in intensity from traditional high-intensity continuous models and is, therefore, more akin to a high-intensity interval model, which is why “super” is added to the name. An empirical study with national-level male team handball players reports that after eight weeks, significant (*p*  <  0.001) improvements with a huge effect size (η^2^p = 0.91) in VO_2_max are obtained [12]. This result is particularly notable when compared to prior studies involving elite athletes [13,14,15]; as such, improvement is achieved during a short cycle of aerobic training. Nevertheless, the question of the effects of prolonged application of the SHCT model remains open.

This study aims to investigate and evaluate the effects of prolonged application of the SHCT model in elite team handball players. We assume that prolonged application of the SHCT training model will lead to improvements in aerobic profile parameters that are equal to or even greater than those achieved with short-term application. This progress should help reach the aerobic capacity levels of top-elite and world-class players. The findings of this study could significantly enhance the aerobic training of female team handball players.

## 2. Materials and Methods

### 2.1. Study Design and Approach to the Research Problem

With the goal of investigating and evaluating the effects of the prolonged application of the SHCT model in elite team handball players, this study applied a field-based quasi-experimental design. This method provided us with high ecological validity and greater external validity. It also offered valuable insights into elite performance relevant to competition through longitudinal monitoring. Additionally, it enabled us to collaborate effectively with coaching staff while ensuring ethical and logistical feasibility. As recommended in a recent review, field-based quasi-experiments with elite athletes struck a critical balance between scientific rigor and real-world applicability [16], making them powerful tools for advancing high-performance sports science.

A time series design with one experimental group, four treatments, one pre-test, three intermediate tests, and one post-test was used to examine the effects of the prolonged application of SHCT on elite female handball players. Between the measurements, the subjects carried out SHCT and all other planned activities. Subjects had SHCT training five times a week during four four-week cycles (sixteen weeks). Sessions were conducted in the team’s training center under the supervision of coaches at the same time of day and under similar environmental conditions. Apart from different aerobic training, there was no difference from the team’s usual type of training. All subjects were assessed on five occasions: 3 days before the first treatment (pre-test), in the middle of the study 3 days after completing each treatment, and 3 days before starting the following four-week sequence of SHCT (intermediate test) as well as at the end of the study 3 days after completing the last SHCT intervention treatment (post-testing). Subjects were motivated and verbally encouraged to give maximum effort in all testing sessions.

Considering that the effects of various aerobic training were topics researched in detail [11,13,14,15,17,18,19,20], the expected effects of other models of designing aerobic training, such as High-Intensity Intermittent Training and High-Intensity Continuous Training, were pretty predictable. That is why comparing the effects obtained in this study with the results of the previous ones could, to some extent, compensate for the lack of a control group.

Also, although the sample size was projected using power analysis of statistical inference due to the specificity of the design, which did not allow sufficient control of confounding variables, and the convenience sampling method was used instead of a random sample, this study should be understood as a team case study. Finally, it should be emphasized that the researchers did not have a direct connection with the participants and the measurers (trainers) when measuring variables and implementing the treatment in question to avoid their influence.

### 2.2. The Participants

Fourteen professional female team handball players (five wingers, three pivots, six backs) competing in the first national handball league of Iceland (Olis Deild Women) participated in the study. Since they have a different training program than the rest of the team, goalkeepers did not participate in the study. Their average age was 22.1 ± 3.4, height 171.8 ± 3.3 cm, weight 72 ± 6.4 kg. The rest of the main characteristics of the sample are presented in Table 1. There were no significant fluctuations in the participants’ body weight during the study period to report. There were also no significant injuries that would exclude any of the participants from the training process for an extended period and, therefore, from the study.

At the beginning of the study, the team was already in the preparation process. The study covered one month before the start of the league as well as three competitive months in which the team played 10 games. At that point in time, the team was lower-middle-ranked in the championship and could be described as an average competition team, which did not stand out in terms of results and abilities compared to other elite teams.

All the subjects were informed about the study aims and procedures and provided written consent for their volunteer participation. The study was conducted in accordance with the European Commission’s General Data Protection Regulation (GDPR), the APA-prescribed ethical principles and code of conduct, and the Declaration of Helsinki. The study design was approved by the Ethical Board (number 484-2) of the Faculty of Sport and Physical Education at the University of Belgrade.

### 2.3. Procedures

Pre-testing involved measurement of age, height, weight, and Cooper’s 12 min run test. Based on this data, a 4-week SHCT was individually designed for each subject. After 4 weeks of SHCT, all the variables were measured again, and a new 4-week SHCT was designed and implemented. Three days after the last aerobic session, final measurements were conducted. All measurement sessions took place at the same time and place and under approximately the same climatic conditions.

### 2.4. Cooper’s 12 min Run Test

Being professional athletes, subjects were familiar with Cooper’s 12 min aerobic run test. They were briefly instructed to run the most significant distance possible on an outdoor track during the 12 min. The Cooper test is widely used for testing athletes and training planning and programming. It has recently shown high validity and reliability with no significant differences compared to a laboratory test [21]. In this study, the test demonstrated high test–retest reliability with Cronbach’s alpha > 0.96, the intraclass correlation coefficient for average measures > 0.96, and a 95% confidence interval ranging from 0.92 to 0.99.

Although laboratory tests are the gold standard in aerobic abilities research, given the aforementioned metric characteristics of Cooper’s test, especially in elite athletes, the choice of field test in this study is more than justified. Also, considering the specific applied goals of this study, the need to improve the aerobic training programming practice, and its similarity with real training and competition situations, field tests are even more appropriate than laboratory ones.

Also, it could be noted that the Cooper test involves continuous running, which does not reflect the intermittent nature of team handball performance, and a sport-specific test or the Yo-Yo Intermittent Recovery Test would be more appropriate for estimating aerobic parameters in the studied population. It should be emphasized that in this study, we are investigating and evaluating the potential of using a continuous model, hence the choice of test. Whether this approach is justified can be concluded based on the results of the study.

Using a handball-specific test could yield valuable insights into the effectiveness of the proposed model concerning the handball game. While this approach is not directly related to the main research question and objectives of this study, it remains a consideration for future research.

### 2.5. Estimates of Aerobic Parameters

For the purposes of designing training and calculating the training effects and changes, age expressed in years, height (BH) expressed in centimeter (cm), weight (BW) expressed in kilograms (kg), and Cooper’s 12 min run test (K) expressed in meters (m) were measured for each subject. From this data, aerobic parameters of VO_2_max expressed in liters per minute (L·min^−1^); VO_2_rel expressed in millimeters per kilogram of body weight in one minute (mL·kg^−1^·min^−1^); velocity at maximal oxygen uptake (vVO_2_max), also known as maximal aerobic speed (MAS) (vVO_2_max), expressed in meters per second (ms^−1^); and muscle efficiency when running in aerobic mode (η) expressed as a percentage (%), were calculated estimated in the following way:VO_2_rel = 3.134304 · 10^−7^ · K^2^ + 0.02077344 · K − 9.03125(1)VO_2_max = [(3.134304 · 10^−7^ · K^2^ + 0.02077344 · K − 9.03125) · BW] · 1000^−1^(2)vVO_2_max = 0.0014 · K + 0.1786(3)η = vVO_2_max/VO_2_max · 5(4)

In addition, the total distance to run in the SHCT cycle (∑DT) expressed in meters (m), total oxygen consumption required for the SHCT cycle (∑VO_2_) expressed in liters (L), and total calorie consumption required for the SHCT cycle (∑kcal) expressed in kilocalories (kcal), were also calculated based on the exercise program.

Aerobic status for each subject was measured and evaluated for the first time 3 days before the start of the first monthly treatment (pre-test). Based on the established condition, training for 4 weeks was programmed for each player (Table 2). Then, the process of measuring and programming was repeated after completing the intermediate test. The last time the aerobic status was evaluated was 3 days after the last monthly treatment (post-testing). Data obtained in initial and intermediate measurements were used for individually programming aerobic training for each subject.

### 2.6. Exercise Program

Since the logic of the program design and redistribution of training loads was the subject of previous studies [11,12], only basic information will be presented here that allows the study to be repeated. Each aerobic training started at 7 or 8 AM. According to the plan, training sessions were to be held on Monday, Tuesday, Wednesday, Friday, and Saturday. Thursday and Sunday were rest days. Uninterrupted continuous aerobic training time was 20 min of running at a speed equal to the percentage of vVO_2_max (Table 2), which was programmed individually for each day, each participant, in the four weeks. The O_2_ consumption for each training session was calculated based on a percentage of VO_2_ max for 20 20-min runs. Cycle average load distribution (oxygen consumption) was 82.75% of VO_2_max. There was one VO_2_ peak in the first week, two in the second, three in the third, and again one peak in the fourth week (Table 2).

All subjects warmed up with five minutes of light running and stretching before each training or testing session and cooled down with ten minutes of light running, relaxation, and stretching after each session.

In addition to 5 morning aerobic training sessions per week, participants had strength and endurance training sessions 2 times per week lasting one hour and also 5 evening special handball training sessions. Of the 5 evenings, 2 were special handball speed training (specific load with the ball with emphasis on the speed of performing tasks), which was held on Mondays and Thursdays. One was special handball problem-solving training (tactical problems dedicated to the actual opponents), which was held on Tuesdays. Moreover, two special handball training sessions were implemented on Wednesdays and Fridays in a light work mode. On Wednesdays, various tasks were carried out that would enable timely and correct answers to the expected match, and on Fridays, tactics for the next match were worked out.

Weather conditions during the study were mainly cold, wet, and windy, with rapid weather changes. The average temperature was between 5 °C and 10 °C, with moderate precipitation.

While running speed is not the most accurate way to estimate VO_2_max during training sessions, we chose not to use heart rate or GPS in our study design to ensure inclusivity. Unfortunately, female team handball remains an underdeveloped sport in many regions around the world. It is not uncommon for even professional senior teams to lack access to the standard technical and material resources that are typically available in men’s sports or other sports.

### 2.7. Statistical Analysis

The sample size was determined after performing a statistical power analysis. For ANOVA: repeated measure, within factors, with an effect size f = 0.25, probability of making a type I error α = 0.05, power 1 − β = 0.90, 1 group, 5 measurements and high expected correlation among measures r = 0.8, the sample size should include at least 12 participants.

The effect of the prolonged application of SHCT on the development of aerobic characteristics of elite female team handball players was evaluated by comparing the training changes and effects obtained from the measurements at various time points. For that purpose, a one-way ANOVA: repeated measure, within-subjects factors, comparing pre-test, three intermediate and post-test measurements of the variables of aerobic status (K, VO_2_max, VO_2_rel, vVO_2_max, η), was used with a chosen level of significance of *p* ≤ 0.001. Bonferroni post-hoc tests were conducted to detect the significance of differences between each measurement session individually. Before applying ANOVA, normality (the Shapiro–Wilk test) and sphericity (Mauchly) tests were conducted on all dependent variables. Cronbach’s alpha and intraclass correlation coefficients were calculated for all dependent variables to assess the consistency and test–retest reliability of data and estimations. As measures of effect, generalized and partial eta squared were reported. For the effect size, partial eta squared was calculated. Criterion for evaluation of the effect size was for η^2^ (0.0001) = very small, η^2^ (0.01) = small, η^2^ (0.06) = medium, η^2^ (0.14) = large, η^2^ (0.26) = very large, and η^2^ (0.5) = huge [22]. Power analyses were performed using G-power 3.1.9.6 (Franz Faul, Universitat Kiel, Germany), while other analyses were conducted with the help of the JAMOVI 1.2.27.0 software package (The jamovi project, 2020. Retrieved from https://www.jamovi.org).

## 3. Results

The descriptive indicators of aerobic status (Table 3) show that at the end of the study, they were improved in the following way: K for 26.8%, VO_2_max for 35.2%, VO_2_rel for 34.9%_,_ vVO_2_max for 25.4%. η for 5.1%, ∑ DT for 26.2%, ∑ VO_2_ for 32.3%, and ∑ kcal for 32.2%.

A one-way within-subjects (repeated measures) ANOVA revealed a significant effect of aerobic training on all the variables of aerobic status. All the obtained effects were huge.

Post hoc tests using the Bonferroni correction revealed that the majority of all increments on all variables of aerobic status, which happened between all tests, were significant at the level of *p* < 0.001. Slightly lower statistical significance levels (*p* = 0.004) were obtained in the second and third measurement of VO_2_rel, first and second measurement (*p* = 0.016), second and fifth measurement (*p* = 0.03) on vVO_2_max, first and second measurement (*p* = 0.004) on η, first and second measurement (*p* = 0.001), third and fourth measurement (*p* = 0.02) of ∑ VO_2_, as well as third and fourth measurement (*p* = 0.02) of ∑ kcal. Non statistically significant (*p* > 0.05) increments were obtained in the second and third, third and fourth, third and fifth, fourth and fifth measurements on vVO_2_max; second and third, fourth, and fifth measurements of η; second and third as well as fourth and fifth measurements of ∑ VO_2_; and second and third as well as fourth and fifth measurements of ∑ kcal. Progress during the study is illustrated in Figure 1 and Figure 2.

There were no significant deviations from the normal distribution or homogeneity assumption violations to report.

## 4. Discussion

This study aimed to investigate and evaluate the effects of prolonged application of the SHCT aerobic training model in elite team handball players. After four months (four cycles lasting four weeks each) of application of SHCT, we obtained significant (*p* < 0.001) improvement in VO_2_max [F (4,13) = 113] with a huge effect size (η^2^p = 0.9), very similar to previous studies when SHCT was applied short-term [12], which is overall in line with our assumptions.

At the beginning of the study (Table 3), although in the middle of preparation for competition, participants had low initial values of aerobic parameters. Not even a single participant, according to the aerobic parameters, belonged to the elite category initially [6]. Of course, the most significant improvement was achieved in the first four weeks of SHCT model implementation. Compared to the results of similar studies [13,14,15], the effect (η^2^ > 0.6) achieved was already significantly higher. After the application of four months of SHCT, all aerobic indicators had a significant (*p* < 0.001) increase with a huge effect size (η^2^ > 0.5). On the other hand, muscle efficiency decreased by 1.2%. The result of the initial measurement can partly explain this result. Participants started with a relatively high mechanical efficiency in contrast to other aerobic indicators. As a result of applying the SHCT model, four handball players reached the world-class zone (VO_2_rel < 57.9), one reached the top-elite zone (VO_2_rel < 55.8), and seven made the elite zone (VO_2_rel < 51.3). A clear positive weather trend is also easily noticeable (Figure 2). The results of muscle efficiency (η) show that handball players have excellent energy efficiency in the motor movement program, as well as the ability to maximize energy mobilization and an outstanding ability to generate muscle force in dynamic aerobic work conditions, which enables them to achieve such results with a proper aerobic training program.

The SHCT model (Table 2) is designed to contribute to enormous training effects (Table 3). Thanks to such program design, oxygen consumption during SHCT training is higher than during matches. The average workload per match was recorded to be 79.4 ± 6.4% of VO_2_max, which is the expenditure for the whole match of 50 L VO_2_ [23,24]. The average load per SHCT training was 82.75% of VO_2_max, which is an expenditure of 64.55 L VO_2_. This result indicates that the demands of matches are 14.55 L of oxygen less than those of SHCT training, representing a 23% decrease. Furthermore, the average demand for oxygen consumption per minute per match was recorded to be 3.49 ± 0.37 L or 49.6 ± 4.8 mL·kg^−1^·min^−1^ [1,4,5,24] while the design of the SHCT model allows the average consumption in one training to be 3.99 ± 0.4 L per minute (more 14.3% VO_2_) or 55.7 ± 3.5 mL·kg^−1^·min^−1^, which is more than 5.7 mL·kg^−1^·min^−1^ every minute. Finally, the design of the SHCT model enables the participants to run an average of 1029.4 m more in the aerobic mode than in the aerobic mode for the entire game [12,25].

The relative value of maximum oxygen consumption, according to laboratory and field tests for the players in national selections, ranges from 45 to 58 mL·kg^−1^·min^−1^, which is higher than the match requirement by 3.6 mL·kg^−1^·min^−1^ [1,4,5]. According to aerobic parameters, the most significant number of national team handball players belongs to the top-elite category, which is influenced by many years of application of traditional aerobic training models. In addition, with traditional aerobic training models, only one to two players per team can be called world-class handball players according to their aerobic profile [6]. The effects of the SHCT model (Table 3 and Figure 1 and Figure 2) are better. After four months of SHCT, four handball players entered the world-class zone, one made it to the top-elite zone, and seven entered the elite zone, while the trend is upward all the time.

The differences in the training effects are significant even if the studies that treated the influence of different types of traditional aerobic training models on the aerobic profile of handball players started from the same initial level as this one [18,26,27]. By applying different aerobic training models, progress was from 4.7 to 6.2% in 8 weeks.

The obtained results are in accordance with previous studies with an SHCT training design [12], where the increase in aerobic variables was significant for the 4 weeks, with effects ranging from 10% to 16%. However, the increase achieved after prolonged application of SHCT in this study was from 25.4% to 35.2% (Table 3).

This study can be criticized for applying continuous training, which does not align with the sport’s demands. Such a claim is, of course, valid, but as the study showed, this approach has a greater effect on general aerobic capacity, which should also have a high transfer to the match itself. The methodological approach allowed us to be in contact with the respondents, who said that they subjectively felt that the training had a significant impact on improving their performance in matches. However, this claim should also be proven by the following empirical study. Also, continuous running is considerably safer regarding injuries compared to intermittent running [28], which involves frequent accelerations and decelerations [29]. This fact raises an important question: Is it worth putting players at risk if both methods produce similar effects on aerobic capacity? While the logical conclusion is that it is not worth the risk, future studies should provide a definitive answer to this question.

Like any other study, this one has limitations that should be taken into account when interpreting the results. In the Method section, we addressed the reasons and advantages of our methodology. In this part, we will focus on shortcomings but with a critical attitude. The lack of a control group, as well as the use of a field test of aerobic capacity, are the most significant limitations of this study. They arose primarily from the practical reasons for the lack of material resources, but also from the specific applied goals of this study. Namely, the primary goal was to test the model on a sample of top athletes. Also, the study had to be conducted during the competitive season, in an extended period, which is why it is very difficult, almost impossible, to find a top team that would serve as a control group. This shortcoming can be partially compensated for with the comparison with previous similar studies, so their samples could be tentatively considered a control, which allows for valid conclusions. The application of field tests is a similar case. Considering that the goal is to apply the results of the study in sports practice, field tests have a greater similarity to the competitive situation than laboratory ones and, therefore, greater construct validity in that particular case. It is also not uncommon for teams to lack the resources to test players frequently enough in the laboratory to program training based on those results.

Furthermore, the methodological design we chose did not allow for proper control of confounding variables, such as the morphological characteristics of the players, playing experience, level of play, workload during matches, and especially the workload of the players in other training situations that were not directly related to the research model. Each of these variables could have influenced the final results. However, given that this is a team of individuals who do not stand out in anything special compared to players of a similar rank and who train according to a routine that is very common for the competition level, it can be assumed that the primary influence on the obtained effects was still the aerobic treatment itself.

In addition to all the mentioned deficiencies, it remains unclear what the side effects of applying the researched model are, which requires implementing a longitudinal study. Considering that the removal of all the mentioned deficiencies requires significant material resources, the results of this study justify their collection.

With all that said in mind, this study primarily opens the door to new research on the design of aerobic training. The proposed model and its effectiveness, as well as side effects, need to be tested on different sports groups according to gender, age, level of competition, and sports branch and discipline. On the other hand, it opens the door for wide use in practice. The study design is very inclusive, and this model of designing aerobic training could be applied without much difficulty by coaches, regardless of whether their clubs have advanced technological and material resources at their disposal. In the end, this method of designing aerobic training is not only practical but also safer than the so far dominant intermittent model, which makes it suitable for application in a wide sports and recreational population.

## 5. Conclusions

The results of this study show that prolonged application of the SHCT model is practical in increasing the parameters of the aerobic profile of elite female team handball players when aerobic testing is performed using a field test. This is why SHCT can be recommended for further use in the aerobic training of female team handball players. Nevertheless, new research on this model inside and outside the context of team handball is needed, but the results of this study also justify it.

## Figures and Tables

**Figure 1 jfmk-10-00241-f001:**
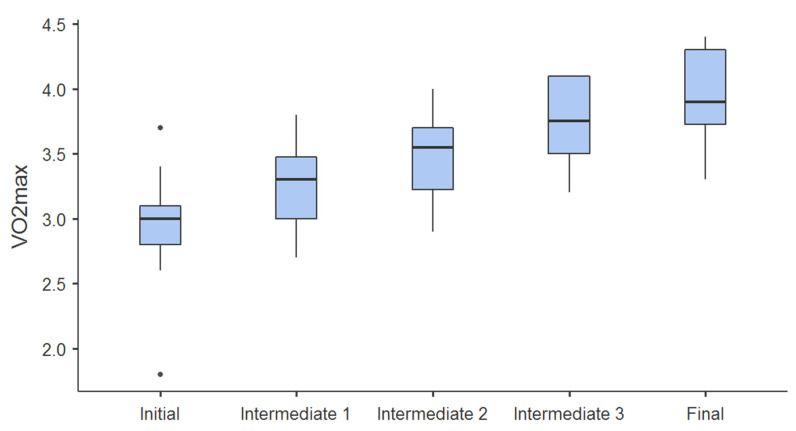
Progress in values of VO_2_max [L·min^−1^] for initial, intermediate, and final measurements.

**Figure 2 jfmk-10-00241-f002:**
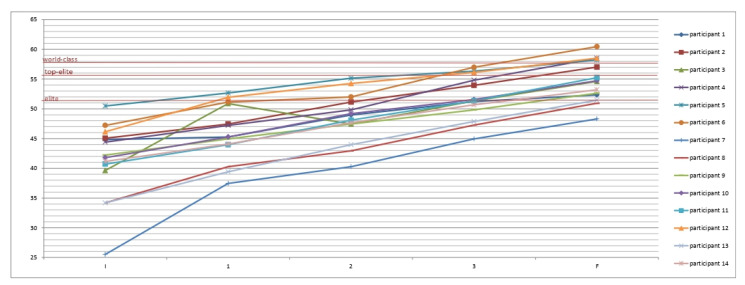
Values of VO_2_rel [mL·kg^−1^·min^−1^] for each participant on initial, intermediate, and final measurements.

**Table 1 jfmk-10-00241-t001:** Description of the Main Characteristics of the Participants.

Participant	Age	BH	BV	VO_2_max	VO_2_rel	vVO_2_max
1	29	173	63	2.81	44.82	3.65
2	20	167.5	60	2.7	45.05	3.68
3	24	175	79	3.13	39.63	3.33
4	28	172	74	3.29	44.4	3.64
5	21	172.5	73	3.68	50.48	4.03
6	19	172	72	3.4	47.22	3.82
7	17	172	72	1.84	25.52	2.42
8	20	174	75	2.57	34.2	2.98
9	20	177	71	3	42.23	3.5
10	22	174	75	3.13	41.8	3.47
11	25	168.5	68	2.77	40.71	3.4
12	20	164	67.5	3.11	46.14	3.75
13	24	172	86	2.94	34.2	2.98
14	21	172	72	2.96	41.15	3.43

Legend: BH—body height expressed in centimeters (cm), BV—body weight expressed in kilograms (kg), VO_2_max—the value of the maximum rate of oxygen consumption expressed in liters per minute (L·min^−1^), VO_2_rel—the value of the maximum relative oxygen consumption expressed in millimeters per kilogram of body weight in one minute (mL·kg^−1^·min^−1^), vVO_2_max—velocity at maximal oxygen uptake expressed in meters per second (ms^−1^).

**Table 2 jfmk-10-00241-t002:** Percentage of VO_2_max and vVO_2_max Used for Daily Aerobic Training Programming.

Week	Monday	Tuesday	Wednesday	Thursday	Friday	Saturday	Sunday
I	85%	75%	95%	Pause	85%	78%	Pause
II	96%	76%	80%	Pause	96%	80%	Pause
III	98%	75%	98%	Pause	75%	98%	Pause
IV	93%	80%	100%	Pause	75%	83%	Pause

**Table 3 jfmk-10-00241-t003:** Descriptive Statistics, ANOVA and Effect Sizes of the Variables of Aerobic.

		K	VO_2_max	VO_2_rel	vVO_2_max	η	∑ DT	∑ VO_2_	∑ kcal
Initial	Mean	2324	2.95	41.3	3.43	23.5	67,685	981	4899
	Median	2360	2.98	42	3.49	22.9	68,599	975	4854
	Standard deviation	295	0.433	6.4	0.412	2.1	8437	146	731
	Minimum	1600	1.84	25.5	2.42	20.7	48,061	615	3077
	Maximum	2750	3.68	50.5	4.03	27.4	80,036	1233	6167
Inter. 1	Median	2657	3.48	48.45	3.90	22.5	77,448	1144	5720
	Standard deviation	190	0.33	4.12	0.27	1.9	5142	101	507
Inter. 2	Median	2808	3.72	51.74	4.11	22	81,602	1226	6128
	Standard deviation	166	0.34	3.60	0.23	2	4618	111	556
Inter. 3	Median	2946	3.93	54.72	4.30	22.3	85,433	1298	6475
	Standard deviation	160	0.37	3.48	0.22	1.9	4442	129	621
Final	Mean	2946	3.93	54.7	4.3	22.3	85,433	1298	6475
	Median	2943	3.89	54.7	4.29	21.9	85,273	1279	6397
	Standard deviation	160	0.373	3.48	0.225	1.9	4442	129	621
	Minimum	2650	3.3	48.3	3.89	19.3	77,256	1103	5515
	Maximum	3210	4.43	60.5	4.67	26.5	92,746	1469	7212
ANOVA	F (4,13)	145 *	113 *	117 *	146 *	13.1 *	147 *	47.1 *	47.6 *
	η^2^p	0.918	0.897	0.9	0.918	0.502	0.919	0.784	0.785

Legend: K—values of distance run in Cooper’s 12 min run test expressed in meters (m), VO_2_max—value of maximum rate of oxygen consumption expressed in liters per minute (L·min^−1^), VO_2_rel—value of the maximum relative oxygen consumption expressed in millimeters per kilogram of body weight in one minute (mL·kg^−1^·min^−1^), vVO_2_max—velocity at maximal oxygen uptake expressed in meters per second (ms^−1^), η—muscle efficiency when running in aerobic mode expressed in percentage (%), ∑ DT—total distance run expressed in meters (m), ∑ VO_2_—total oxygen consumption expressed in liters (L), ∑ kcal—total calories consumed (kcal), F-statistics with df (2,26), * *p* < 0.001, η^2^p—partial eta squared.

## Data Availability

All data are available at https://docs.google.com/spreadsheets/d/1jwiGRyvP3yxjc8yNqa5rw4dFBJWN5oKx03rDNaCAxck/edit?gid=0#gid=0 (accessed on 1 March 2025).

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
