# Peer review of "Effectiveness of Prolonged Application of Super High-Intensity Continuous Training—Team Case Study"

_jfmk, 2025, doi:10.3390/jfmk10030241_

Round 1
Reviewer 1 Report
Comments and Suggestions for Authors
Dear authors,
Congratulations on excellent work and effort. Taking into account that it is a quasi-experimental design, it can be accepted that the claims are convincing, clearly written, well presented, supported with data and discussed properly.
However, I have a few minor suggestions and one major concern about the results of the effects.
A) When abbreviations are mentioned for the first time, put the full name of the mentioned parameter:
Maximum relative oxygen consumption (VO2rel)
Velocity at maximal oxygen uptake (vVO2max) also known as maximal aerobic speed (MAS)
B) Add Table: Descriptive statistics of the main characteristics of the participants.
Descriptive statistics of the main characteristics of the participants should be presented more clearly.
It should be presented in Table 1. in the results section (age, body weight, height, VO2max etc). If you have more sample features it should be presented in Table 1. Sample size is relatively small, so quality of sample description is also important. Through that table, the readers' experience will be better. If you have not collected such data, ignore this suggestion.
C) 2.1. Study Design and Approach to the Research Problem and also 2.3. Procedures
“A time series design with one experimental group, four treatments, and one pre-test, three intermediate, and one post-testing was used to examine the effects of the prolonged application of SHCT on elite female handball players.”
“Pre-testing involved measurement of age, height, weight, and Cooper's 12-minute run test. Based on this data, a 4-week SHCT was individually designed for each subject. After 4 weeks of SHCT, all the variables were measured again, and a new 4-week SHCT was designed and implemented. Three days after the last aerobic session, final measurements were conducted.”
- Results of three intermediate testing and measures are not presented in paper. It should be presented and it should be part of statistical analysis (T1 ,T2, T3, T4, T5).
Major concern: Without a control group, it becomes difficult to isolate the effects of the experimental treatment and distinguish them from other factors. A control group helps validate the experiment by providing a benchmark against which to measure the effects of the treatment.
Overall review: The quality of the sample is elite but relatively small. A treatment that lasts 16 weeks, while the training is individually programmed again every four weeks, certainly adds quality to this research. Although more variables were derived and statistically processed, in essence only one test was performed. Since it is a quasi-experimental research (without a control group), I think it would have been much better if the authors had statistically processed all 5 instances of tests and measures in order to see and analyze and understand the differences throughout the entire competitive season (when was the highest peak in results, when was there stagnation, etc). Therefore, this research could be improved but still I find it very usefull.
Author Response
Reviver 1
Dear authors,
Congratulations on excellent work and effort. Taking into account that it is a quasi-experimental design, it can be accepted that the claims are convincing, clearly written, well presented, supported with data and discussed properly.
Response: Thank you very much for your time, effort and support.
However, I have a few minor suggestions and one major concern about the results of the effects.
Response: We truly believe in the peer review process, and we do our best to implement all of your recommendations.
- A) When abbreviations are mentioned for the first time, put the full name of the mentioned parameter:
Maximum relative oxygen consumption (VO2rel)
Velocity at maximal oxygen uptake (vVO2max) also known as maximal aerobic speed (MAS)
Response: we did that on pages 1 and 4 and in tables
- B) Add Table: Descriptive statistics of the main characteristics of the participants.
Descriptive statistics of the main characteristics of the participants should be presented more clearly.
It should be presented in Table 1. in the results section (age, body weight, height, VO2max etc). If you have more sample features it should be presented in Table 1. Sample size is relatively small, so quality of sample description is also important. Through that table, the readers' experience will be better. If you have not collected such data, ignore this suggestion.
Response: we add Table 1 with Description of the Main Characteristics of the Participants
- C) 2.1. Study Design and Approach to the Research Problem and also 2.3. Procedures
“A time series design with one experimental group, four treatments, and one pre-test, three intermediate, and one post-testing was used to examine the effects of the prolonged application of SHCT on elite female handball players.”
“Pre-testing involved measurement of age, height, weight, and Cooper's 12-minute run test. Based on this data, a 4-week SHCT was individually designed for each subject. After 4 weeks of SHCT, all the variables were measured again, and a new 4-week SHCT was designed and implemented. Three days after the last aerobic session, final measurements were conducted.”
- Results of three intermediate testing and measures are not presented in paper. It should be presented and it should be part of statistical analysis (T1 ,T2, T3, T4, T5).
Response: We add results to table 3. Analyses are presented in Result section: Post hoc tests using the Bonferroni correction revealed that the majority of all increments on all variables of aerobic status, which happened between all tests, were significant at the level of p < 0.001. Slightly lower statistical significance levels (p = 0.004) were obtained in the second and third measurement of VO2rel, first and second (p = .016), second and fifth measurement (p = 0.03) on vVO2max, first and second measurement (p = 0.004) on η, first and second (p = 0.001), third and fourth measurement (p = 0.02) of ∑ VO2, as well as third and fourth measurement (p = 0.02) of ∑ kcal. Non statistically significant (p > 0.05) increments were obtained in the second and third, third and fourth, third and fifth, fourth and fifth measurement on vVO2max, second and third, fourth, fifth measurements of η, second and third as well as fourth and fifth measurement of ∑ VO2, second and third as well as fourth and fifth measurement of ∑ kcal.
Major concern: Without a control group, it becomes difficult to isolate the effects of the experimental treatment and distinguish them from other factors. A control group helps validate the experiment by providing a benchmark against which to measure the effects of the treatment.
Response: This is of course true, and we stated this in the limitations of the study while in the method we tried to explain why we chose this design.
Overall review: The quality of the sample is elite but relatively small. A treatment that lasts 16 weeks, while the training is individually programmed again every four weeks, certainly adds quality to this research. Although more variables were derived and statistically processed, in essence only one test was performed. Since it is a quasi-experimental research (without a control group), I think it would have been much better if the authors had statistically processed all 5 instances of tests and measures in order to see and analyze and understand the differences throughout the entire competitive season (when was the highest peak in results, when was there stagnation, etc).
Response: we add results in table 1, process the data in results section and add figure 1 to illustrate progress during study.
Therefore, this research could be improved but still I find it very usefull.
Response: We think that this text is significantly improved thanx to your comments. Kind regards

Reviewer 2 Report
Comments and Suggestions for Authors
The authors studied a dozensized handball team during 4 months of intense physical training and found continously improving oxygen capacity in all the players and studied as a group.
The authors acknowledge the possible biases in subject selection, person charactedristics and majorly lack of cointrol group.
Moreover, the authors state that these effects are well known (thus no control group).
The authors could elaborate a little more on the additional value of the study,
Author Response
Reviver 2
The authors studied a dozensized handball team during 4 months of intense physical training and found continously improving oxygen capacity in all the players and studied as a group.
The authors acknowledge the possible biases in subject selection, person charactedristics and majorly lack of cointrol group.
Response: we agree
Moreover, the authors state that these effects are well known (thus no control group).
Response: We clarified it: Having in mind that the effects of various aerobic training are topics researched in detail [11,13–15,17–20], expected effects of other models of designing aerobic training such as High-Intensity Intermittent Training, High-Intensity Continuous Training are pretty predictable, that is why comparing the effects obtained in this study with the results of the previous ones can to some extent compensate for the lack of a control group.
The authors could elaborate a little more on the additional value of the study,
Response: We did that in the end of discussion: With all that said in mind, this study primarily opens the door to new research on the design of aerobic training. The proposed model and its effectiveness, as well as side effects, need to be tested on different sports groups according to gender, age, level of competition, and sports branch and discipline. On the other hand, it opens the door for wide use in practice. The study design is very inclusive, and this model of designing aerobic training could be applied without much difficulty by coaches regardless of whether their clubs have advanced technological and material resources at their disposal. In the end, this method of designing aerobic training is not only practical but also safer than the so far dominant intermittent model, which makes it suitable for application in a wide sports and recreational population.
THANK YOU VERY MUCH FOR YOUR TIME EFFORT AND COMMENTS.

Reviewer 3 Report
Comments and Suggestions for Authors
You did a great job with your study. Here are a few suggestions and questions I would like you to address.
Abstract:
Can you provide an example of a load for SHCT?
“Recent studies have shown significant positive effects, but it remains unknown what effects its prolonged use has.” is awkward. Please rewrite for better clarity and smoother flow.
“which opens up the perspective of using the SHCT model...” is vague, can you add more detail of how it can be used?
Introduction:
The first sentence is long and somewhat cumbersome.
Repetitive mention of "training models", please edit.
Please maintain tense consistency.
Methods:
The text shifts between past and present tense in this section
Limitations of the methodology should be held to the Discussion section
What were the climactic conditions for testing?
Was there resistance/anaerobic training performed during the study?
Is using running speed the best way to estimate VO2Max during training sessions, why not use heart rate or GPS?
Why did you not perform a handball specific test to the study?
Results:
Good presentation
Discussion
How does your results compare to other studies following the first 4 weeks of training?
Conclusion:
Please add when aerobic testing is performed using a field test.
Author Response
Reviver 3
You did a great job with your study. Here are a few suggestions and questions I would like you to address.
Response: Thank you very much for your time, effort and support. We believe in the peer review process, and we do our best to implement all of your recommendations.
Abstract:
Can you provide an example of a load for SHCT?
Response: we did that: Super High-Intensity Continuous Training (SHCT) is a type of aerobic training program that combines high intensity with continuous loads, such as running for 20 minutes at 75%, 80%, or even 95% of the velocity at maximal oxygen uptake.
“Recent studies have shown significant positive effects, but it remains unknown what effects its prolonged use has.” is awkward. Please rewrite for better clarity and smoother flow.
Response: we rewrite it: Response: Recent studies show significant positive effects, but the consequences of prolonged use remain unknown.
“which opens up the perspective of using the SHCT model...” is vague, can you add more detail of how it can be used?
Response: we rewrite it: The investigated aerobic model is effective. Therefore, its use is recommended for designing aerobic training for elite teams and the general sports population.
Introduction:
The first sentence is long and somewhat cumbersome.
Response: we rewrite it: The modern female team handball game demands that players endure alternating aerobic and anaerobic loads. This is often accompanied by shorter and longer breaks, with frequent, sudden, and rapid changes in both intensity and direction of movement during gameplay [1–3].
Repetitive mention of "training models", please edit.
Response: we edited it: The ability of the players to fulfil the demands of the games at the top international level, which constantly require new and more efficient training models, has increased in recent years [7,8]. The aim is to improve aerobic characteristics as well.
Several models exist for designing aerobic training, which can be classified into two categories: continuous and interval training. Continuous training enhances capillarization and the oxidative capacity of muscles [9]. In contrast, interval training primarily improves the strength of the heart muscles by incorporating varying intensities, which boosts circulation and enhances the delivery of oxygen to the muscles [10]. The objective of both models is to increase aerobic capacity by positively influencing both the organic and metabolic systems. With the aim to meet growing demands, Super High-Intensity Continuous Training (SHCT), which combines high intensity with continuous load, was proposed [11]. The model significantly differs in intensity from traditional high-intensity continuous models and is, therefore, more akin to a high-intensity interval model, which is why "super" is added to the name. In an empirical study with national-level male team handball players, after eight weeks, significant (P < 0.001) improvements with extraordinarily large effect size (η2p=0.91) in VO2max were obtained [12]. This result is particularly notable when compared to prior studies involving elite athletes [13–15], as such improvement was achieved during a short cycle of aerobic training. Nevertheless, the question of the effects of prolonged application of the SHCT model remained open.
This study aims to investigate and evaluate the effects of prolonged application of the SHCT model in elite team handball players. We assume that prolonged application of the SHCT training model will lead to improvements in aerobic profile parameters that are equal to or even greater than those achieved with short-term application. This progress should help reach the aerobic capacity levels of top elite and world-class players. The findings of this study could significantly enhance the aerobic training of female team handball players.
Please maintain tense consistency.
Response: we did that also in whole introduction: The modern female team handball game demands that players endure alternating aerobic and anaerobic loads. This is often accompanied by shorter and longer breaks, with frequent, sudden, and rapid changes in both intensity and direction of movement during gameplay [1–3]. In such a structure, aerobic effort prevails during matches [4,5]. Namely, at the top competitive level, the average load is 79.4 ± 6.4% of the maximum rate of oxygen consumption (VO2max) for a full game [4]. At the same time, the results of studies with laboratory and field tests show that the maximum relative oxygen consumption (VO2rel) of the players from various national teams ranges from 45 to 58 ml. kg-1. min-1 [5]. A recent study that analyzes team handball on-court physical performance in elite, top-elite, and world-class female players finds significant differences among groups in aerobic parameters (Elite VO2rel = 54.2 ± 2.9 ml. kg-1. min-1; Top-elite VO2rel = 60.6 ± 4.8 ml. kg-1. min-1; and World-class VO2rel = 64.3 ± 6.4 ml. kg-1. min-1) [6]. The ability of players to fulfill the demands of the game at the top international level, which constantly requires new and more efficient training models, has increased in recent years [7,8]. The aim is to improve aerobic characteristics as well.
Several models exist for designing aerobic training, which can be classified into two categories: continuous and interval training. Continuous training enhances capillarization and the oxidative capacity of muscles [9]. In contrast, interval training primarily improves the strength of the heart muscles by incorporating varying intensities, which boosts circulation and enhances the delivery of oxygen to the muscles [10]. The objective of both models is to increase aerobic capacity by positively influencing both the organic and metabolic systems. With the aim to meet growing demands, Super High-Intensity Continuous Training (SHCT), which combines high intensity with continuous load, is proposed [11]. This model significantly differs in intensity from traditional high-intensity continuous models and is, therefore, more akin to a high-intensity interval model, which is why "super" is added to the name. An empirical study with national-level male team handball players reports that after eight weeks, significant (P < 0.001) improvements with huge effect size (η2p=0.91) in VO2max are obtained [12]. This result is particularly notable when compared to prior studies involving elite athletes [13–15], as such improvement is achieved during a short cycle of aerobic training. Nevertheless, the question of the effects of prolonged application of the SHCT model remains open.
This study aims to investigate and evaluate the effects of prolonged application of the SHCT model in elite team handball players. We assume that prolonged application of the SHCT training model will lead to improvements in aerobic profile parameters that are equal to or even greater than those achieved with short-term application. This progress should help reach the aerobic capacity levels of top elite and world-class players. The findings of this study could significantly enhance the aerobic training of female team handball players.
Methods:
The text shifts between past and present tense in this section
Response: We edit it to the past tense: With the goal of investigating and evaluating the effects of the prolonged application of the SHCT model in elite team handball players, this study applied a field-based quasi-experiment design. This method provided us with high ecological validity and greater external validity. It also offered valuable insights into elite performance relevant to competition through longitudinal monitoring. Additionally, it enabled us to collaborate effectively with coaching staff while ensuring ethical and logistical feasibility. As recommended in a recent review, field-based quasi-experiments with elite athletes struck a critical balance between scientific rigor and real-world applicability [16], making them powerful tools for advancing high-performance sports science.
A time series design with one experimental group, four treatments, one pre-test, three intermediate tests, and one post-test was used to examine the effects of the prolonged application of SHCT on elite female handball players. Between the measurements, the subjects carried out SHCT and all other planned activities. Subjects had SHCT training five times a week during four four-week cycles (sixteen weeks). Sessions were conducted in the team's training center under the supervision of coaches at the same time of day and under similar environmental conditions. Apart from different aerobic training, there was no difference from the team's usual type of training. All subjects were assessed on five occasions: 3 days before the first treatment (pre-test), in the middle of the study 3 days after completing each treatment, and 3 days before starting the following four-week sequence of SHCT (intermediate test) as well as at the end of the study 3 days after completing the last SHCT intervention treatment (post-testing). Subjects were motivated and verbally encouraged to give maximum effort in all testing sessions.
Having in mind that the effects of various aerobic training were topics researched in detail [11,13–15,17–20], expected effects of other models of designing aerobic training such as High-Intensity Intermittent Training and High-Intensity Continuous Training were pretty predictable. That is why comparing the effects obtained in this study with the results of the previous ones could, to some extent, compensate for the lack of a control group.
Also, although the sample size was projected using power analysis of statistical inference, due to the specificity of the design, which did not allow sufficient control of confounding variables, and the convenience sampling method instead of a random sample, this study should be understood as a team case study. Finally, it should be emphasized that the researchers did not have a direct connection with the participants and the measurers (trainers) when measuring variables and implementing the treatment in question to avoid their influence.
Limitations of the methodology should be held to the Discussion section
Response: We delete the sentence: We chose this approach despite its noticeable shortcomings, which we will analyze in detail during the discussion.
What were the climactic conditions for testing?
Response: We add information: Weather conditions during the study were mainly cold, wet and windy, with rapid weather changes. The average temperature was between 5–10°C with moderate precipitation.
Was there resistance/anaerobic training performed during the study?
Response: We add information: participants have strength and endurance training session 2 times per week lasting one hour
Is using running speed the best way to estimate VO2Max during training sessions, why not use heart rate or GPS?
Response: We add explanation: While running speed is not the most accurate way to estimate VO2max during training sessions, we chose not to use heart rate or GPS in our study design to ensure inclusivity. Unfortunately, female team handball remains an underdeveloped sport in many regions around the world. It is not uncommon for even professional senior teams to lack access to the standard technical and material resources that are typically available in men's sports or other sports.
Why did you not perform a handball specific test to the study?
Response: We add explanation: Using a handball-specific test could yield valuable insights into the effectiveness of the proposed model concerning the handball game. While this approach is not directly related to the main research question and objectives of this study, it remains a consideration for future research.
Results:
Good presentation
Thanx
Discussion
How does your results compare to other studies following the first 4 weeks of training?
Response: We add information: Of course, the most significant improvement was achieved in the first four weeks of SHCT model implementation. Compared to the results of similar studies [13–15], the effect (η2 > 0.6) achieved was already significantly higher.
Conclusion:
Please add when aerobic testing is performed using a field test.
Response: We add this in conclusion: The results of this study show that prolonged application of the SHCT model is practical in increasing the parameters of the aerobic profile of elite female team handball players when aerobic testing is performed using a field test.
THANK YOU VERY MUCH FOR YOUR TIME EFFORT AND COMMENTS.
